# Metabolic and Cardiorespiratory Responses of Semiprofessional Football Players in Repeated Ajax Shuttle Tests and Curved Sprint Tests, and Their Relationship with Football Match Play

**DOI:** 10.3390/ijerph17217745

**Published:** 2020-10-23

**Authors:** Tomasz Gabrys, Arkadiusz Stanula, Urszula Szmatlan-Gabrys, Michal Garnys, Luboš Charvát, Subir Gupta

**Affiliations:** 1Department of Physical Education and Sport Science, Faculty of Pedagogy, University of West Bohemia, 301 00 Pilsen, Czech Republic; tomaszek1960@o2.pl (T.G.); lcharvat@ktv.zcu.cz (L.C.); 2Institute of Sport Science, The Jerzy Kukuczka Academy of Physical Education, Mikołowska 72A, 40-065 Katowice, Poland; 3Department of Anatomy, Faculty of Rehabilitation, University of Physical Education, 31-571 Krakow, Poland; urszula.szmatlan@awf.krakow.pl; 44Sport Lab, Laboratory of Physical Preparation, 00-951 Warsaw, Poland; m.garnys@4sportlab.pl; 5Faculty of Medical Sciences, University of West Indies, Cave Hill 11000, Barbados; s.gupta@awf.katowice.pl

**Keywords:** aerobic, performance decrement, fatigue, lactate, football

## Abstract

In this study, the Ajax Shuttle Test (AST) and the Curved Sprint Test (CST) were conducted on semiprofessional football players to evaluate (1) their test performance, (2) the extent of anaerobic glycolysis by measuring blood lactate, (3) performance decrement and onset of fatigue, and (4) the correlation between selected physiological variables and test performance. Thirty-two semiprofessional Polish football players participated in this study. Both AST and CST were conducted on an outdoor football ground and were conducted in two sets; each set had six repetitions. In the case of AST, the total duration for 6 repetitions of the exercise in Sets 1 and 2 were 90.63 ± 3.71 and 91.65 ± 4.24 s, respectively, whereas, in the case of CST, the respective values were 46.8 ± 0.56 and 47.2 ± 0.66 s. Peak blood lactate concentration [La] after Sets 1 and 2 of AST were 14.47 ± 3.77 and 15.00 ± 1.85 mmol/L, and in the case of CST, the values were 8.17 ± 1.32 and 9.78 ± 1.35 mmol/L, respectively. Performance decrement in AST was more than in CST, both after Set 1 (4.32 ± 1.43 and 3.31 ± 0.96 in AST and CST, respectively) and Set 2 (7.95 ± 3.24 and 3.71 ± 1.02 in AST and CST, respectively). Only in a few of the repetitions, pulmonary ventilation (V_E_) and oxygen uptake (VO_2_) were found to be significantly correlated with the performance of the volunteers in both AST and CST. Respiratory exchange ratio (RER) was significantly correlated with most of the repetitions of AST, but not with CST. The study concludes that (1) AST shows more dependence on the anaerobic glycolytic system than shorter repetitive sprints (as in CST), (2) there is more performance decrement and fatigue in AST than in CST, and (3) early decrease in performance and fatigue in the semiprofessional football players in AST and CST may be due to the insufficiency of their aerobic energy system.

## 1. Introduction

The work-rate profile and variations in movement and activity patterns of football players during competitive matches demand the engagement of all the energy-yielding systems. The relative contribution of the aerobic and anaerobic systems in football games largely depends on the level of competition and the training status, motor conditioning, and genetic factors of the individual [1,2,3]. Efficacy of the aerobic system in football players determines their ability to carry out activities of repeated maximal and submaximal intensities, e.g., cruising, sprinting, and jumping, besides game-specific activities like dribbling and sliding [4,5]. Two important factors determine the energy cost of players during play—the way they move on the pitch and the way they lead the ball. The energy cost of running forward is lower than the energy expenditure involved in running backwards and sidewards [6,7].

Progression of a football match is usually associated with a gradual decrease in the speed of runs, a decrease in the number of sprints, and a shortening of distance covered with maximum speed [8,9]. A rapid fall of glycogen in the working muscles of the players with the advancement of the match has been indicated as the culprit of such slowing of speed and frequency of sprints [10]. A decrease in the blood lactate of the footballers, in the later part of matches, also supports this view [11].

The mobility of football players during the game is partly dependent on anaerobic glycolysis [12,13]. Intense exercise, sustaining for longer than 20 s, is associated with an exponential rise in blood lactate accumulation [14]. Fatigue resulting from various activities during a football match, with short rest pauses in between, gradually increases the time of carrying out further activities as the match progresses, especially in the later part of the game. Faster removal of lactate from the blood with the progression of the match is an important issue because it, at least partly, decides how quickly the player will be fatigued. The training of footballers is also focused on this important aspect of faster removal of lactate by using lactate as a potential source of energy. A number of studies [15,16,17,18,19] have concluded that the level of aerobic efficiency determines the limits of the intensity of work. Oxygen insufficiency in working muscle is considered to be a regulatory factor of the metabolic changes that cause the exhaustion of muscle glycogen resources and the inhibition of the activity of glycolytic enzymes. These two are considered to be the major factors resulting in the decrease of the efficiency of ATP synthesis and, hence, the appearance of signs of muscle fatigue [10].

Playing football, regardless of the sports level, is based on moving with high intensity, which is separated by intervals of low and medium intensity activities [20,21,22]. Accelerations in a match, lasting from 2–4 s [23], occur many times in succession, often not separated by a clear break [22,24,25]. The importance of intense efforts is given by the fact that they are the ones that determine the result in a match [26,27]. In football, equally important, and often more important, is the ability to accelerate nonlinearly [28]. About 85% of maximum speed maneuvers consist of curvilinear sprints of varying curvature [24,29,30]. The curves of the run usually have a radius of 3.5–11 m and are performed by players in all positions, except for goalkeepers [30]. The assessment of this area of the player’s preparation, with particular emphasis on playing in conditions of increasing fatigue, is important in planning the training process. ASTs and SCTs require multiple direction changes, simulating the movement of a soccer player during the game. On the other hand, multiple-use makes it possible to bring the subject’s assessment closer to the conditions of the game, during which this effort is performed repeatedly. The game of football is characterized by a wide range of movements, and it is nearly impossible to mimic the movement of football players during match play in any laboratory or field tests. The Ajax Shuttle Test (AST) and the Curved Sprint Test (CST) are designed to imitate some of the very basic movement patterns of football players on the ground [31,32]. Yo-Yo Intermittent Recovery Test Level 1 (YoYo IR1), on the other hand, is a nonspecific test that measures the ability to perform high intensity intermittent aerobic work, as in football [33]. This study aims to compare some relevant physiological parameters and fatigue development in semiprofessional football players in two types of repeated sprint tests—AST and CST. Both the sprint tests (AST and CST) conducted in this study involved a change of direction on similar surfaces but differ in distance and duration of the run that likely to differ their metabolic responses. This study also evaluates the endurance capacity of the players and investigates if any predictable relationship exists between indicators of endurance capacity and their performance in sprint tests.

## 2. Materials and Methods

### 2.1. Subjects

Thirty-two semiprofessional, male football players were selected for this study. Regular active participation (≥75% of the team’s total time of match play in the most recent season) in the 3rd or 4th Division Polish football league and regular participation in training sessions (at least 5 times in a week) were the two most important selection criteria for the volunteers. The participants had trained for 10 ± 4 years. Age, height, and bodyweight of the volunteers were 24.5 ± 2.8 years, 176 ± 3.1 cm, and 76.3 ± 3.6 kg, respectively. Subjects were instructed not to eat for at least three hours before testing and to maintain normal dietary habits for two days before testing. All the tests were conducted at least 48 h after competitive match play or a heavy training session, if any. The study was approved by the Bioethical Commission at the District Medical Chamber in Krakow.

### 2.2. Main Test Procedures

The whole study was conducted in three major steps: Yo-Yo Intermittent Recovery Test Level 1 (YoYo-IR1), Curved Sprint Test (CST), and Ajax Shuttle Test (AST). All the tests were conducted on a standard-sized outdoor football ground and under similar ambient conditions. The anaerobic efficiency of the volunteers was assessed by CST and AST, whereas YoYo-IR1 was conducted to assess their aerobic fitness. Both CST and AST were preceded by a 20-min warm-up, including stretching, jogging, and sprinting. Warm-up in both the cases was ended 5 min before the main tests. The research was conducted in the following order: Saturday—match; Sunday-free; Monday—Yo-Yo Test; Tuesday—technical and tactical training; Wednesday—Curved Sprint Test; Thursday—45 min start-up; Friday—Ajax Shuttle Test.

Yo-Yo Intermittent Recovery Test Level 1 (YoYo-IR1). This test is a widely used field test to measure the ability to perform aerobic work intermittently, especially by recreational and semiprofessional athletes [34]. YoYo-IR1 triggers a nearly maximal stimulation of the aerobic system. On a 20-m course, participants run out and back with a 10-s active break after each 40-m (i.e., 20 m × 2). Speed increases after each bout of 20-m × 2 until they are unable to continue. The speed of each bout is controlled by an audio recorder. All subjects were familiarized with the test by at least one pretest.

The purpose of this test is to predict VO_2max_ and measure variables like pulmonary ventilation (V_E_), heart rate (HR), respiratory frequency (RF), respiratory exchange ratio (RER), and oxygen pulse (VO_2_/beat) at the most exhaustive stage of the test (that likely indicates the attainment of the VO_2max_).

Curved Sprint Test. This consists of running a distance of 30-m with maximum speed 6 times (or 6 repetitions) and changing the direction of the run 3 times in each repetition. A 25-s break is given after each repetition, during which the subject has to return to the start line before the next repetition begins (Figure 1). The first set (Set 1) of CST consists of a 30-m distance, repeated 6 times, followed by another set (Set 2) of CST, having identical protocol. The time gap between the two sets of CST is 6 min, where the volunteers are on 2 min of passive rest, followed by 2 min of jogging and 2 min of passive rest again. The time required to complete all the repetitions in both the sets is recorded.

Ajax Shuttle Test. This test consists of running a 40-m distance 6 times (repetitions), keeping the speed as fast as possible. The direction of the run has to change 4 times in each repetition. The subject has a 25-s break between two consecutive repetitions, during which he goes back to the start line (Figure 2). Like CST, AST is completed in 2 sets—Set 1 and Set 2—with a 6-min break between the two.

Collection of blood samples and measurement of blood lactate in CST and AST. Capillarized blood samples were collected from the fingertips of the participants for the measurement of blood lactate. The time of collection of blood samples, in both CST and AST, was just before and after the end of both Sets 1 and 2, at the 3rd minute of the recovery period after both Sets 1 and 2, and at the 6th and 9th minute of the recovery period after Set 2. The lactate level in blood was determined by the enzymatic amperometric method (Biosen S_line, EKF-Diagnostics GmbH, Barleben, Germany).

Recording of heart rate. The heart rate (HR) of the volunteers, in all the cases of field tests, was recorded continuously by a heart rate monitor (Polar Team Pro, Kempele, Finland) at an interval of 5 s.

Other physiological parameters like oxygen consumption (VO_2_), pulmonary ventilation (V_E_), respiratory frequency (RF), heart rate (HR), respiratory exchange ratio (RER), and oxygen pulse (VO_2_/beat) were recorded by a portable K4b2 metabolic analyzer (Cosmed, Rome, Italy). YoYo-IR1 field tests were conducted in this study [35,36]. Volunteers selected a rubber facemask (Hans-Rudolph, Kansas City, MO, USA) of a suitable size. The K4b2 system was warmed up for 40 min prior to calibration, in accordance with the manufacturer’s guidelines. The weight of the Cosmed K4b2 system is 1.5 kg, including the battery and a specially designed harness. This system is highly reliable for both laboratory and field tests [37,38]. The calibration involves 10 pumps of a 3-L syringe into the Cosmed turbine for the volume of expired air during ventilation, a room air calibration (20.93% O_2_ and 0.03% CO_2_), and a calibration with a standard gas mixture of O_2_ (16%) and CO_2_ (5%). The values of V_E_, RF, VCO_2_, and VO_2_ were measured continuously.

Determination of performance decrement index (PDI) from sprint results in CST and AST. Performance decrement index is determined from the difference between the actual time of run for a given distance and the hypothetical shortest time to cover the distance [39]:
PDI = 100 × [(TT/IT) − 1]

where IT (ideal time) is the best sprint time, usually the first sprint time of the six repetitions of Set 1 of CST or AST. Thus, IT (in seconds) = FST × 6, where FST is the best (shortest) sprint time among the 6 repetitions of Set 1. Total time (TT) is the sum of the 6 repetitions of sprints in CST and AST. Thus, TT = ST1 + ST2 + ST3 + ST4 + ST5 + ST6, where ST1 through ST6 are the sprint times from the 1st to the 6th repetitions of Set 1 (in CST and AST). Time was recorded using the SMARTSPEED PRO time gate system (Fusion Sport, Brisbane, Australia), with an accuracy of 0.01 s.

Determination of fatigue index (FI) from sprint results. Fatigue index is a measure of the drop-off in performance [39] and is calculated from the following equation:

FI = 100 × [(MST − FST)/FST]

where MST and FST refer to the mean sprint time and fastest (or best) sprint time, respectively.

### 2.3. Statistical Analysis

Means and standard deviations were used to represent the average and typical spread of values of all the performance variables of the participants. The normal Gaussian distribution of the data was verified by the Shapiro–Wilk test. A paired-sample *t*-test was used to detect differences between Sets 1 and 2 in each test (AST and CST). The Bonferroni correction for all multiple comparisons was applied. This takes α = 0.05/K, where K is the number of comparisons. Hence, with K = 6, the result of a comparison is considered significant if *p* < 0.008. The relationship between PD and IF, obtained from the AST and CST scores, and between the physiological parameters of the Yo-Yo tests were determined with Pearson’s product-moment correlation analysis. The level of significance was set at 0.05 for all the remaining tests. Calculations were performed with the Statistica 13.3 statistical software package (TIBCO Software Inc., Palo Alto, CA, USA).

## 3. Results

Predicted maximal oxygen uptake in both mL/min and mL/kg/min, RF, V_E_, HR, and oxygen pulse recorded at the predicted VO_2max_ level are presented in Table 1.

The analysis of the data obtained was carried out in three steps. In the first step, the time differences between corresponding repetitions of any given set (Set 1 or 2) of AST and CST were determined. Any difference in PDI and FI for a given set between CST and AST was determined in the second step. Correlations between the physiological variables and the scores (time) of each repetition of AST and CST were determined in the third step. The duration of all repetitions in Sets 1 and 2 of AST are presented in Table 2. The total duration of Sets 1 and 2 of AST was 90.63 ± 3.71 and 91.65 ± 4.24 s, respectively, and their difference (1.02 s or 1.1%) was statistically nonsignificant. Only in the case of the 6th repetition, the difference between Sets 1 and 2 was found significant. For any given repetition, the speed was faster in Set 1 than Set 2. There was a gradual decline in speed in both the sets from Repetition 1 through 6.

Table 3 shows the mean durations of the two sets of CST. The duration of any particular repetition of a set in CST is almost half the duration of the corresponding repetition in AST. However, the trend of change in the duration of the repetitions in CST is very similar to AST. The total duration of Set 1 (46.8 ± 0.56) was significantly lower than Set 2 (47.2 ± 0.66).

Blood lactate levels [La] at rest and after each set of AST and CST are presented in Table 4. In the case of AST, [La] after Set 2 (15.00 ± 1.85) was higher than after Set 1 (14.47 ± 3.77), although the difference was nonsignificant. However, Set 2 [La] were found significantly higher than Set 1 [La] of CST. Blood lactate concentrations, after both sets of AST, were found to be significantly higher than CST.

The PDI and FI values, after both sets of AST and CST, are presented in Table 5. In AST, both PDI and FI were higher after Set 2 than Set 1, although the difference was statistically significant in the case of PDI only. The trend was similar in the case of CST. However, FI after Set 2 was much higher (291.52%) than Set 1, and the difference was highly significant (*p* < 0.001).

The correlations between time and physiological variables during AST are presented in Table 6. A significant correlation between V_E_ and time was noted only in the first repetition of Set 2. In a few instances (Repetition 1 of Set 1 and Repetitions 1 and 5 of Set 2), the correlation between VO_2_ and time was statistically relevant. Significant negative correlations in a number of repetitions of Set 1 were noted between RER and time, although no such correlation was significant in Set 2. A positive correlation between HR and time was noted, although no such correlation was statistically significant except for the first repetition of Set 1. The only statistically significant correlations existing between oxygen pulse and time were found in the last repetition of Set 1 and the first repetition of Set 2 of CST.

Correlations between the physiological variables and time in all the repetitions of CST are shown in Table 7. In most of the repetitions of CST, strong correlations between V_E_ and time were found. Similarly, VO_2_ was found to be significantly correlated with time in four of the repetitions. Correlations between RER and time were negative in most of the instances but statistically significant in the third and last repetitions of Set 2 only. Time vs. HR correlation was noted to be always positive and was highly significant in most of the repetitions. Oxygen pulse was significantly correlated with time only in the first repetition of both Sets 1 and 2.

Correlations between the physiological variables and PDI and FI are given in Table 8. No regular pattern of correlations was observed between these physiological variables and PDI and FI. However, HR was found significantly correlated only with FI_2_ (FI in Set 2) and oxygen pulse with FI_2_.

## 4. Discussion

The aerobic capacity of the football players in this study, as indicated by predicted VO_2max_ (ranging from 46.96 to 59.74 mL/kg/min, with the mean value of 53.54 mL/kg/min), is comparable to that of second and third division Polish League football players, as reported by researchers [40,41,42,43]. Reduction of running efficiency during a match is a phenomenon that very often affects the outcome of the sports match and is the result of fatigue [44,45,46]. The economy of movement is a determinant of endurance, which is a complex result of many functions, including metabolic and cardiopulmonary functions [45,47]. Metabolic efficiency relates to the use of available energy, while cardiorespiratory efficiency relates to the supply and use of oxygen in the working muscles. It is therefore related to heart rate, pulmonary ventilation, and VO_2max_. In football, which is a sport with a long duration and variable intensity, movement economy is a relevant parameter of aerobic fitness [48]. The credibility of the assessment of running economics requires the use of tests based on sport-specific movement patterns [49]. The classic approach based on running sections in a straight line is not informative for football [17,18,50]. It is known that changes in direction are significantly more stressful than linear runs, causing higher HR and blood lactate [51,52,53,54]. Thus, the footballers of the present study constitute a representative sample for this sports level. Professional football players carry out 1200 to 1400 different activities, of which 700 to 800 are movements requiring changes in direction by sprinting, jumping, stopping, or reaccelerating. Only 10–12% of the movements that are carried out maximally have a significant impact on the match result [8].

Tests (AST and CST) conducted in this study involved high-intensity running in different directions, with a slight but steady decline in speed. A relatively longer pause can partly restore speed, as observed in this study—for example, speed in the first repetition of Set 2 was faster than the last repetition of Set 1 for both AST and CST. A longer pause between intense activities can partly restore energy sources. However, recovery is only partial when such intermittent activities continue. This can explain why the speed of a football match slowly declines with the advancement of time, especially in the later part of the second half of the game [27]. Highly trained football players with better running economy are able to play with relatively higher speed and less fatigue than semiprofessional players [55]. Like real match play, an efficient physiological response in AST and CST depends on the appropriate training and running economy of the players. The dominant role of the aerobic energy system in football games has been suggested by many researchers [15,17]. High VO_2max_ allows athletes to run intermittently, with brief pauses, more efficiently than athletes with low aerobic capacity. Higher aerobic capacity promotes energy replenishment during intensive intermittent workouts, typical of football matches and training [4], thus reducing fatigue by sparing glycogen and preventing lactate accumulation in muscle and blood [10,56].

In semiprofessional football players, high-intensity workouts in the first half or early stages of the match lead to reduced [La] in the later part of the game, but, in the case of more professional footballers, this may not always be the case [16,19,57].

Very fast but longer durations (14 to 16 s) of runs, repeated 6 times with short breaks (25 s) between the two repetitions, lead to very high lactate concentration after each set of AST. Shorter breaks did not allow the lactate to be removed sufficiently by oxidation but triggered lactate production. A six-minute pause between the sets was not enough for significant removal of lactate, and, as a result, the lactate production-to-removal ratio was higher and the lactate level further increased after Set 2. A rise in lactate concentration in CST can be explained in the same way. However, the shorter duration of each sprint (7.55 to 8.07 s) was unable to stimulate the anaerobic glycolysis completely, and the energy was adequately supplied by the phosphagen system.

Researchers [58,59] have reported that in real football match play, [La] is highest around the middle of the game and declines as the match approaches its end. The [La] mostly varies between 6 and 12 mmol/L, which is closer to the [La] in CST but is much less than [La] in AST. Low-intensity movements (e.g., jogging, walking, cruising) between sprints in football matches allow lactate to be oxidized and does not allow [La] to rise very high. Quick repetitions of longer sprints for 14 s or longer are not very common in football, and this can explain the relatively low [La] during football match play than in AST. The research of Bradley et al. [60] on the Premier League showed that the recovery time between repetitions of high-intensity workouts increases. Between the 1st and the 6th 15-min episodes of the game, the cooldown time was 28% longer regardless of the position. Not only was a reduction in the number of accelerations observed, but there was also a reduction in the frequency of attacks. Thus, the effectiveness of the game largely depends on the speed of regeneration during the match and the ability to work at maximum intensity throughout the match. This may support training to increase aerobic capacity, as Buchheit et al. [61] pointed out in their research on handball players. Training at the VO_2max_ level can be effective in this respect by using intermittent efforts [62,63,64]. The results of Helgerud et al. [65] showed that the use of high-intensity intermittent exercise in training may result in greater improvement in maximum aerobic capacity and endurance than continuous exercise at an intensity of 85% of the maximum heart rate.

The physiological profiles of the semiprofessional football players participating in this study were not homogenous, and, thus, the variables recorded in AST and CST showed a significant relationship, at least in some cases, between VO_2max_ and performances in AST and CST [4]. Nyberg et al. [66] showed that in highly trained footballers, a period of speed endurance training enhanced the performance of intermittent high-intensity work. Krustrup et al. [33] indicated that there is a relationship between the aerobic power of football players, the level of the team, and the total distance covered by the football players during match play. Football is a multitask sport, which requires an optimum blending of physical, technical, and tactical abilities in order to make a player efficient [6,7]. Multidirectional movements, jumps, dribbles, tackles, and slides require complex interaction between both aerobic and anaerobic energy systems.

The scope of engaging aerobic metabolism in ongoing submaximal activities depends on the efficiency of the aerobic system and the restoration of muscle glycogen during the breaks [6,40]. During repeated, high-intensity efforts, as was the case with AST and CST, up to 40% of energy may come from aerobic metabolism [67]. Test results indicate that aerobic metabolism in semiprofessional football players does not complement anaerobic metabolism from the moment of beginning work of submaximal and maximal intensities. The methods of the study meet the criteria required to assess a football player in conditions corresponding to those of a football game. In order to avoid the influence of surface–footwear interaction on power generation [68], both the surface and footwear used by the participants were preserved in the current study. The assessment used the multidirectional intermittent movement carried out by the football players because the participation of aerobic metabolism can depend on the neuromechanical factor of greater muscle activity [69] and reduced joint stiffness, resulting in a decrease in the use of elastic energy [70] when accelerating, slowing down, and changing directions in comparison with running at a constant speed with little change in direction (e.g., a run on a treadmill). The metabolic cost of running on a track or a treadmill is different from the movement of players on a football ground, which demands multidirectional runs with pauses in between, plus jumps, quick postural changes, and rapid changes in running intensity. The nature of the movement involved in AST and CST conditions is comparable to real or simulated football match play and involves higher oxygen and energy usage and more dependence on the glycolytic pathway than running forward and in a line, as on a treadmill. This can be the cause of the differences among the relationships recorded in the present study and studies by other researchers [7,71]. Further studies should concentrate on devising a methodology of training that would increase the efficiency of work in a group of semiprofessional football players.

A reciprocal relationship between RER and test performance, in both AST and CST, suggests insufficient buffering of the lactate produced in the muscles during the workout periods. This probably reflects the fact that the training of the semiprofessional football players does not develop the aerobic system significantly for the removal of lactate from muscle and blood. Dolci et al. [48] concluded that movement economy is one of the relevant physiological parameters of football players, and aerobic fitness plays a key role in improving movement economy, which is an important prerequisite in high-level competitive football matches.

## 5. Conclusions

The study concludes that (1) AST shows more dependence on the anaerobic glycolytic system than shorter repetitive sprints (as in CST), (2) performance decrement and fatigue are more in AST than in CST, and (3) early decrease in performance and fatigue in semiprofessional football players in AST and CST may be due to the insufficiency of their aerobic energy system.

## Figures and Tables

**Figure 1 ijerph-17-07745-f001:**
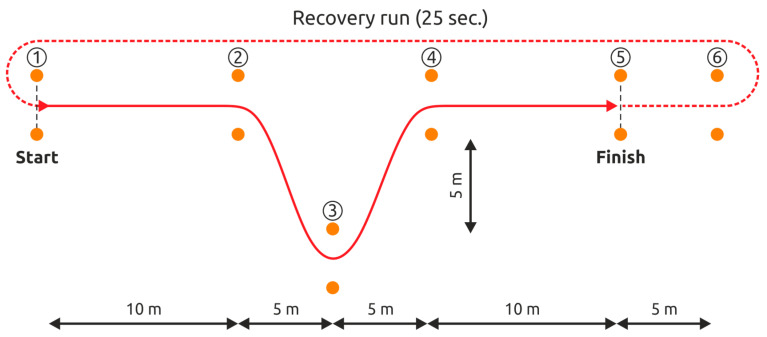
The Curved Sprint Test.

**Figure 2 ijerph-17-07745-f002:**
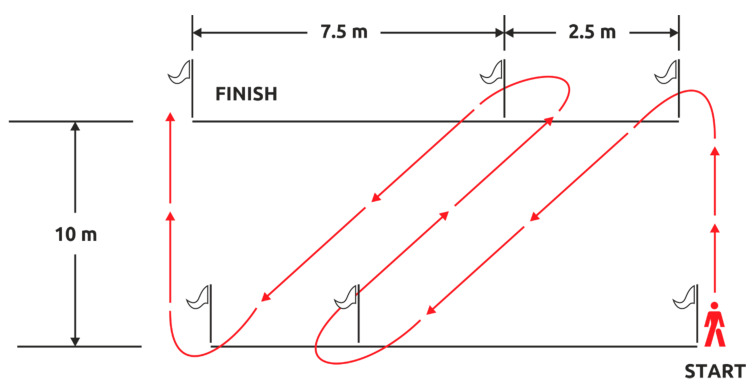
The Ajax Shuttle Test.

**Table 1 ijerph-17-07745-t001:** Predicted VO_2max_ and some of the physiological variables at the level of the predicted VO_2max_ of the volunteers.

Variable	Mean ± SD	Range
VO_2max_ (mL/kg/min)	53.5 ± 2.4	47.0–59.7
RF (b/min)	59.5 ± 2.8	48.3–67.4
V_E_ (L/min)	128.8 ± 9.4	100.5–147.3
HR (bpm)	189.1 ± 3.2	177.0–199.0
Oxygen pulse (mL/beat)	21.4 ± 1.4	17.3–25.7

**Table 2 ijerph-17-07745-t002:** Time scores in Sets 1 and 2 of the Ajax Shuttle Test (AST) and their differences.

Repetition	Set 1 (sec)	Set 2 (sec)	Differences (%)	*p-*Value
1	14.49 ± 0.81	14.64 ± 1.04	0.16 (1.0)	0.187
2	14.89 ± 0.83	15.00 ± 1.04	0.12 (0.7)	0.338
3	15.14 ± 0.83	15.21 ± 0.94	0.07 (0.5)	0.607
4	15.33 ± 0.89	15.42 ± 1.00	0.09 (0.6)	0.540
5	15.45 ± 0.88	15.63 ± 1.08	0.18 (1.2)	0.165
6	15.35 ± 0.91	15.76 ± 1.05	0.41 (2.6)	0.001
Total time	90.63 ± 3.71	91.65 ± 4.24	1.02 (1.1)	0.127

**Table 3 ijerph-17-07745-t003:** Time scores in Sets 1 and 2 of the Curved Sprint Test (CST) and their differences.

Repetition	Set 1 (sec)	Set 2 (sec)	Differences (%)	*p*-Value
1	7.55 ± 0.09	7.59 ± 0.14	0.04 (0.5)	0.019
2	7.74 ± 0.11	7.77 ± 0.12	0.03 (0.4)	0.169
3	7.87 ± 0.17	7.88 ± 0.16	0.01 (0.1)	0.494
4	7.90 ± 0.11	7.91 ± 0.13	0.01 (0.1)	0.815
5	7.89 ± 0.13	7.98 ± 0.13	0.10 (1.1)	<0.001
6	7.86 ± 0.09	8.07 ± 0.17	0.21 (2.7)	<0.001
Total time	46.8 ± 0.56	47.2 ± 0.66	0.39 (0.8)	<0.001

**Table 4 ijerph-17-07745-t004:** Blood lactate concentrations (mmol/L) at rest and after * AST and CST.

Field Tests	Blood Lactate Concentrations (mmol/L)			Differences (%)	*p*-Value
	Rest	Set 1	Set 2		
Ajax Shuttle Test	1.38 ± 0.46	14.47 ± 1.86	15.00 ± 1.85	0.52 (3.7)	0.410
Curved Sprint Test	1.43 ± 0.51	8.17 ± 1.32	9.78 ± 1.35	1.60 (19.7)	<0.001
Differences (%)	–0.05 (–3.6%)	6.3 (43.5%)	5.22 (34.8%)		
*p*-value	n.s.	<0.001	<0.001		

* Only the highest postexercise blood levels were considered.

**Table 5 ijerph-17-07745-t005:** Performance decrement index (PDI) and fatigue index (FI) in CST and AST.

Test/Parameter	Set 1	Set 2	Differences (%)	*p*-Value
PDI in AST	4.32 ± 1.43	7.95 ± 3.24	–3.63 (–45.7)	<0.001
FI in AST	6.25 ± 2.57	7.81 ± 3.37	–1.56 (–20.0)	0.062
PDI in CST	3.31 ± 0.96	3.71 ± 1.02	–0.40 (–12.1)	0.014
FI in CST	1.65 ± 0.70	6.46 ± 2.08	–4.81 (–291.5)	<0.001

**Table 6 ijerph-17-07745-t006:** Correlations between some physiological variables at the predicted VO_2max_ level and time required for various repetitions of AST and CST.

Physiological Parameters	Repetition in Series in Ajax Shuttle Test Set 1						Repetition in Series in Ajax Shuttle Test Set 2					
	1	2	3	4	5	6	1	2	3	4	5	6
V_E_ (L/min)	0.29	0.12	0.17	0.10	0.17	0.10	0.38 *	0.16	0.19	0.16	0.14	0.14
VO_2_ (mL/min)	0.35 *	0.18	0.17	0.28	0.25	0.34	0.44 ^†^	0.03	0.09	0.10	0.31 *	−0.01
RER (VCO_2_/VO_2_)	−0.03	−0.17	−0.35 *	−0.32 *	−0.32 *	−0.38 *	−0.16	−0.16	−0.05	−0.24	−0.21	−0.13
HR (bpm)	0.32 *	0.30	0.22	0.29	0.29	0.17	0.30	0.14	0.21	0.25	0.22	0.11
VO_2_/HR (mL/bpm)	0.27	0.10	0.12	0.20	0.17	0.32 *	0.36 *	−0.02	0.02	0.03	0.27	−0.05

* *p* ≤ 0.05; ^†^
*p* ≤ 0.01.

**Table 7 ijerph-17-07745-t007:** Correlation coefficients between the time of repetitions of CST and physiological parameters at the predicted VO_2max_ level in the Yo-Yo Test.

Physiological Parameters	Repetition in Series in Curved Sprint Test 1						Repetition in Series in Curved Sprint Test 2					
	1	2	3	4	5	6	1	2	3	4	5	6
V_E_ (L/min)	0.49 ^†^	0.27	0.33 *	0.41 ^†^	0.33 *	0.22	0.44 ^†^	0.43 ^†^	0.41 ^†^	0.44 ^†^	0.47 ^†^	0.35 *
VO_2_ (mL/min)	0.44 ^†^	0.03	0.10	0.11	0.31 *	0.00	0.35 *	0.19	0.18	0.28	0.25	0.35 *
RER (VCO_2_/VO_2_)	−0.14	−0.19	−0.04	−0.17	−0.08	−0.19	0.09	−0.01	−0.31 *	−0.12	−0.18	−0.44 ^†^
HR (bpm)	0.67 ^‡^	0.44 ^†^	0.41 ^†^	0.50 ^†^	0.30	0.42 ^†^	0.37 *	0.34 *	0.59 ^‡^	0.44 ^†^	0.49 ^†^	0.65 ^‡^
VO_2_/HR (mL/bpm)	0.34 *	−0.07	0.01	0.01	0.29	−0.11	0.32 *	0.14	0.07	0.22	0.18	0.24

* *p* ≤ 0.05; ^†^
*p* ≤ 0.01; ^‡^
*p* ≤ 0.001.

**Table 8 ijerph-17-07745-t008:** Relationship between PD and FI in AST and CST and physiological parameters at the predicted VO_2max_ level during the Yo-Yo Test.

Physiological Parameters	Ajax Shuttle Test				Curved Sprint Test			
	PDI_1_	PDI_2_	FI_1_	FI_2_	PDI_1_	PDI_2_	FI_1_	FI_2_
V_E_ (L/min)	0.02	0.04	0.10	0.08	−0.08	0.12	−0.19	0.13
VO_2_ (mL/min)	−0.02	−0.10	−0.01	0.08	−0.30	0.12	−0.08	0.30
RER (VCO_2_/VO_2_)	0.23	−0.13	0.11	−0.28	0.00	−0.15	0.07	−0.23
HR (bpm)	−0.17	0.13	−0.02	0.28	−0.15	0.20	−0.22	0.34 *
VO_2_/HR (mL/bpm)	0.02	−0.14	0.00	0.02	−0.31 *	0.08	−0.04	0.25

* *p* ≤ 0.05; PDI_1_ and PDI_2_ are PDIs in Sets 1 and 2, respectively, and so are FI_1_ and FI_2_.

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
