# Peer review of "Metabolic and Cardiorespiratory Responses of Semiprofessional Football Players in Repeated Ajax Shuttle Tests and Curved Sprint Tests, and Their Relationship with Football Match Play"

_ijerph, 2020, doi:10.3390/ijerph17217745_

Round 1
Reviewer 1 Report
Good afternoon, I liked the work a lot. It complies with all the rules of a research work with all the sections very well differentiated and explained. If I have to raise any question, it is related to the comparisons between professional and semi-professional players. The work is of sufficient quality to focus only on that level. It is clear that the parameters of the professionals will be above by multiple and varied aspects.
Congratulations on the job.
Author Response
Comments 1
Good afternoon, I liked the work a lot. It complies with all the rules of a research work with all the sections very well differentiated and explained. If I have to raise any question, it is related to the comparisons between professional and semi-professional players. The work is of sufficient quality to focus only on that level. It is clear that the parameters of the professionals will be above by multiple and varied aspects.
Congratulations on the job.
Response 1
We thank the reviewer for the time and effort in reviewing this manuscript and appreciate the positive feedback.

Reviewer 2 Report
The authors have done an interesting job. They seek to evaluate the relationship between cardiovascular and respiratory response parameters (measured with a portable gas analyzer) of soccer players in intermittent exercise (Yo-Yo IR1) and performance in two field tests determined by anaerobic glycolysis: The Ajax Shuttle Test (AST) and a Curved Sprint Test (CST). It is especially interesting to look for field tests that evaluate performance dependent on anaerobic glycolysis since they are not very frequent and because, precisely, anaerobic glycolysis is the energy route that worsens most with detraining (eg off-season or injured). However, in my opinion, there are methodological approaches that must be justified two or corrected.
The authors have used a discontinuous incremental test such as Yo-Yo IR1 to evaluate VO2max or VEmax. Given that in Yo-Yo IR1 the aerobic load approaches the maximum, but the anaerobic system may be a limiting factor (Krustrup et al 2003 - PMID: 12673156), it would have been a better choice an incremental ramp test that depends only on VO2max (so, allowing to measure it). We do not know if this is the reason why the authors write VO2 and VE in the Yo-Yo test (instead of VEmax or VO2max).
A ramp protocol is not only interesting in terms of isolating the aerobic potential from other performance limitations in the test itself, but it also allows to better carry out part of the study's objectives: to study the relationships between VO2max and VEmax and intermittent discontinuous performance in the field (AST and CST). To see the influence of VO2max, VO2 kinetics or VO2 recovery on repeated sprint performance, it is better (at least more frequently in the literature) to record them in transitions to constant speeds, expressed as % of V-Vo2max or maximal aerobic speed (MAS) (that are obtained in incremental ramp protocols). As examples: Dupont et al 2005 - PMID: 15976999, Dupont et al 2010 - PMID_20574678 or Christensen et al 2011 - PMID: 21311360. For this reason, the authors should justify why the Yo-yoIR1 choice.
I am not sure I understand the role of the RER and the importance of its correlation with other parameters. The RER needs at least 2 - 3 min of constant exercise to give an approximation of the aerobic or anaerobic participation. In a discontinuous incremental exercise, in addition to energy participation, the VE capacity, progressive desaturation of O2 and CO2 through sprints/recoveries, blood PCO2, the sensitivity of pH receptors, etc. also influence.
I have searched the information about the validity and repeatability AST and CST in PubMed (references 20 and 21 provided do not give them) and I have not been able to find them (I apologize). Reading the article, I'm not sure if the CST is already validated or if the authors propose it. In the case of what the authors propose: did they perform measurements to calculate inter-session repeatability and intra-session CV?
Line 34 and 93: anaerobic (instead of anerobic).
Author Response
Comments and Suggestions for Authors
Comments 1
The authors have done an interesting job. They seek to evaluate the relationship between cardiovascular and respiratory response parameters (measured with a portable gas analyzer) of soccer players in intermittent exercise (Yo-Yo IR1) and performance in two field tests determined by anaerobic glycolysis: The Ajax Shuttle Test (AST) and a Curved Sprint Test (CST). It is especially interesting to look for field tests that evaluate performance dependent on anaerobic glycolysis since they are not very frequent and because, precisely, anaerobic glycolysis is the energy route that worsens most with detraining (eg off-season or injured).
Response 1
We thank the reviewer for the time and effort in reviewing this manuscript and appreciate the constructive feedback. Below we have answered the comments and suggestions from the reviewer. We have marked changes in the text in red.
Comments 2
The authors have used a discontinuous incremental test such as Yo-Yo IR1 to evaluate VO2max or VEmax. Given that in Yo-Yo IR1 the aerobic load approaches the maximum, but the anaerobic system may be a limiting factor (Krustrup et al 2003 - PMID: 12673156), it would have been a better choice an incremental ramp test that depends only on VO2max (so, allowing to measure it). We do not know if this is the reason why the authors write VO2 and VE in the Yo-Yo test (instead of VEmax or VO2max).
A ramp protocol is not only interesting in terms of isolating the aerobic potential from other performance limitations in the test itself, but it also allows to better carry out part of the study's objectives: to study the relationships between VO2max and VEmax and intermittent discontinuous performance in the field (AST and CST). To see the influence of VO2max, VO2 kinetics or VO2 recovery on repeated sprint performance, it is better (at least more frequently in the literature) to record them in transitions to constant speeds, expressed as % of V-Vo2max or maximal aerobic speed (MAS) (that are obtained in incremental ramp protocols). As examples: Dupont et al 2005 - PMID: 15976999, Dupont et al 2010 - PMID_20574678 or Christensen et al 2011 - PMID: 21311360. For this reason, the authors should justify why the Yo-yoIR1 choice.
Response 2
The choice of the Yo-Yo IR 1 test was not accidental. It is a test commonly used to assess aerobic capacity in soccer players. The structure of the test much more closely corresponds to the effort of a football player than to the continuous test. Of course, there is a limitation in the athlete's assessment related to the anaerobic component. Therefore, direct measurement of VO2, CO2 and VE was used.
It should also be remembered that this limitation also applies to continuous testing. However, it is not as exposed as in intermittent tests, where usually higher maximum speeds are achieved.
Comments 3
I am not sure I understand the role of the RER and the importance of its correlation with other parameters. The RER needs at least 2 - 3 min of constant exercise to give an approximation of the aerobic or anaerobic participation. In a discontinuous incremental exercise, in addition to energy participation, the VE capacity, progressive desaturation of O2 and CO2 through sprints/recoveries, blood PCO2, the sensitivity of pH receptors, etc. also influence.
Response 3
We fully agree with the reviewer's opinion. The RER value always increases during a break after a strenuous exercise while VE decreases. In our case, only RER values recorded during the effort were used for the analysis. The aim was to determine the increase in exhaled CO2 as a result of increased activation of anaerobic metabolism.
Comments 4
I have searched the information about the validity and repeatability AST and CST in PubMed (references 20 and 21 provided do not give them) and I have not been able to find them (I apologize). Reading the article, I'm not sure if the CST is already validated or if the authors propose it. In the case of what the authors propose: did they perform measurements to calculate inter-session repeatability and intra-session CV?
Response 4
The tests were selected to meet the following criteria: they were described in the literature, the working time was clearly different, they were used in football, they were known to the group of respondents from previous experiences. In this way, the phenomenon of learning movement was avoided. The amount of work performed was different due to the effort time. The tests were not intended to evaluate the subjects during longer periods of training sessions (mesocycle, preparation period or competition period). Their goal was to make an effort with a structure of movement similar to that repeated many times during a match. Perhaps the test term is not the happiest. However, it was suggested by the name used in the football training literature. The tests were not validated by us, because they performed the task of exerting effort, and not the assessment of the anaerobic capacity of the players.
Comments 5
Line 34 and 93: anaerobic (instead of anerobic).
Response 5:
We thank the reviewer for this suggestion and have revised this term throughout the paper
Reviewer 3 Report
Metabolic and Cardiorespiratory Responses of Semi-Professional Football Players in Repeated Ajax Shuttle Test and Curved Sprint Test, and their Relationship with Football Match Play
The study is of interest and has been well executed. The number of subjects is sufficient, although no power analysis has been performed. The manuscript suffers from, at times, poor English and should be submitted to a thorough copyediting.
The different parts of the manuscript are mostly accurately phrased, but at places sentences can be improved, see later. Some of the references seems not to be suitable for the point the authors want to make: Ref. 11 does not cover football, but uphill running. The references 16,19 and 36 does not seem to be good choices. Please do check and read in full length all your references.
Line 75: The sentence describing the aim of the study is long and unclear. Should be broken down and made clearer. Do you want to assess fatigue mechanisms? Role of the glycolytic system in what? You probably mean correlate a number of…
In the Methods, you should be more accurate in the description of subject selection. Did You select the subjects? Is this a convenience sample, or could the subjects choose to take part?
Were the three tests conducted in a set order, or where they counter balanced?
What is “anaerobic efficiency”, be as clear as possible with the descriptions
During the Yo-Yo-IR1, did the subjects wear the K4b2? If so, report the weight of the system
Cosmed srl, Rome, Italy
The manufacturer’s calibration guidelines recommend using two different calibration gases (air and something different from that).
Since there were six repetitions to compare, a Bonferroni correction should be applied to the p-level
Results
Table 1: One significant digit is probably enough
Table 2: How was the time taken, with what accuracy? State this in the Methods part
Table 6: use two digits in all the numbers. Would not R2 be more informative here and in other tables?
Correlations does not indicate causality, so you should comment the results in Table 7 and 8 more in this respect. Also use the same description, “correlation” or “relationship” in the table text.
Minor issues:
Line 255 has grammatical errors.
Line 284 is long and unclear, what is meant by “complement the anaerobic…”?
Line 289: is “preserved” a good word here? Can “identical” be used?
Line 289-294: Long sentence, needs to be rephrased.
Line 294: “… metabolic cost of running…”
Line 295: “…is different from movements…”
The Conclusion should be precise. “…continue for some time…”
Author Response
Comments and Suggestions for Authors
Metabolic and Cardiorespiratory Responses of Semi-Professional Football Players in Repeated Ajax Shuttle Test and Curved Sprint Test, and their Relationship with Football Match Play
Comment 1
The study is of interest and has been well executed. The number of subjects is sufficient, although no power analysis has been performed. The manuscript suffers from, at times, poor English and should be submitted to a thorough copyediting.
Response 1:
We thank the reviewer for the time and effort in reviewing this manuscript and appreciate the constructive feedback. Below we have answered the comments and suggestions from the reviewer. We have marked the changes in the text in blue
Comment 2
The different parts of the manuscript are mostly accurately phrased, but at places sentences can be improved, see later. Some of the references seems not to be suitable for the point the authors want to make: Ref. 11 does not cover football, but uphill running. The references 16,19 and 36 does not seem to be good choices. Please do check and read in full length all your references.
Response 2
Yes that's right, a very valuable comment. The choice of this source was due to the fact that running in football with frequent changes of direction requires good strength preparation. Such preparation is achieved in training during ascents. Of course, without prejudice to the manuscript, we can remove this reference from list of references.
Comment 3
Line 75: The sentence describing the aim of the study is long and unclear. Should be broken down and made clearer. Do you want to assess fatigue mechanisms? Role of the glycolytic system in what? You probably mean correlate a number of…
Response 3
Thank you for paying attention, we have modified the research objective in the following way:
“This study was aimed to compare some relevant physiological parameters and fatigue development in semiprofessional football players in two types of repeated sprint tests – AST and CST. Both the sprint tests (AST and CST) conducted in this study involved change of direction on similar surface but differ in distance and duration of run that likely to differ their metabolic responses. This study also evaluated endurance capacity of the players and investigated if any predictable relationship exists between indicators of endurance capacity and their performance in sprint tests”.
Comment 4
In the Methods, you should be more accurate in the description of subject selection. Did You select the subjects? Is this a convenience sample, or could the subjects choose to take part?
Response 4
The subjects were players of one team. They trained according to the same organizational system. The same training methods were used in their preparation. They were aware of the research being conducted. The research results were used by the trainers to rationalize the training activities. Each subject could refuse to participate in the research or stop participating in the research at any time. The criterion for selecting the respondents was systematic participation in training (5 times a week), training internship for a minimum of 6 years and participation in the local football league.
Comment 5
Were the three tests conducted in a set order, or where they counter balanced?
Response 5
We have supplemented the Methods section with the following explanation:
“The research was conducted in the following order: Saturday - match; Sunday - free; Monday - Yo-Yo Test; Tuesday - technical and tactical training; Wednesday - Curved Sprint Test; Thursday - 45 min start-up; Friday - Ajax Shuttle test”.
Comment 6
What is “anaerobic efficiency”, be as clear as possible with the descriptions
Response 6
Anaerobic capacity (the maximum power of aerobic metabolism) is essential for sub- and maximal intensity movements. In these movements, the total amount of energy is used that combines the energy output from ATP (adenosine triphosphate), PC (phosphocreatine) and glycolysis (lactic acid system). These conditions require maximum aerobic and anaerobic performance. Therefore, aerobic and anaerobic fitness contribute to the critical values as the dominant performance factor in team games.
Comment 7
During the Yo-Yo-IR1, did the subjects wear the K4b2? If so, report the weight of the system Cosmed srl, Rome, Italy
Response 7
The weight of the system is approximately 1.5 kg.
Comment 8
The manufacturer’s calibration guidelines recommend using two different calibration gases (air and something different from that).
Response 8
Each of the respondents had an individual rubber facemask selected (Hans-Rudolph, Kansas City, MO) which was placed over the participants' mouth and nose. The Cosmed K4b2 system was warmed up for 40 minutes prior to calibration in accordance with the manufacturer’s guidelines. The calibration involved 10 pumps of a 3 L syringe into the Cosmed turbine for volume of expired air during ventilation, a room air calibration (20.93% O2 and 0.03% CO2), and a calibration with a standard gas mixture of O2 (16%), and CO2 (5%).
Comment 9
Since there were six repetitions to compare, a Bonferroni correction should be applied to the p-level
Response 9
We applied the Bonferroni correction for multiple comparisons
Comment 10
Results
Table 1: One significant digit is probably enough
Response 10
We thank the reviewer for this suggestion and have corrected data in the table 1
Comment 11
Table 2: How was the time taken, with what accuracy? State this in the Methods part
Response 11
Time was recorded using SMARTSPEED PRO time gate system (Fusion Sport, USA) with accuracy of 0.01 s.
Comment 12
Table 6: use two digits in all the numbers. Would not R2 be more informative here and in other tables?
Response 12
We thank the reviewer for this suggestion and have corrected data in the table 6.
We also decided to keep the correlation coefficients
Comment 13
Correlations does not indicate causality, so you should comment the results in Table 7 and 8 more in this respect.
Also use the same description, “correlation” or “relationship” in the table text.
Response 13
We thank the reviewer for this suggestion and have corrected
Comment 14
Minor issues:
Line 255 has grammatical errors.
Line 284 is long and unclear, what is meant by “complement the anaerobic…”?
Line 289: is “preserved” a good word here? Can “identical” be used?
Line 289-294: Long sentence, needs to be rephrased.
Line 294: “… metabolic cost of running…”
Line 295: “…is different from movements…”
Response
We thank the reviewer for bringing this to our attention. We have carefully proof-read the revised manuscript to eliminate grammatical errors.

Reviewer 4 Report
The work presented in this manuscript examined the performance of semi-professional football players in AST and CST. The authors concluded that the AST was more dependent on the anaerobic glycolytic system than CST, more performance decrement and fatigue were found in AST that CST, and the players were not heavily dependent on aerobic system that caused early decrease in performance and fatigue.
Although to examine the physical fitness profile of football players and develop optimal testing and training methods are important, and this study may have a merit if that was case, the manuscript as presented does not effectively achieve this goal and some of the conclusions do not appear to be supported by the data collected in this study. I would provide the following comments and critiques for authors’ consideration, aiming to improve the quality of the presentation.
- The rationale and aim of the study must be better justified.
As the authors stated in Lines 70-72, it is very difficult to mimic the movement of football players (not to mention their specific positions) during a football match play by laboratory or field tests. Therefore, it would be better to provide more detailed background information, in physiological terms, about the tests to be examined. E.g., what are previous findings in respect the general sprint and low intensity movement durations, and evidence of fatigue or rate of fatigue, during a football match, and whether/how the tests used in this study could simulate the football profile (or not). The authors appeared to trying to explain the physiological/metabolic characteristics of football players in the Introduction, but failed to effectively link these to the tests to be used. If the aim was to assess fatigue in these tests, it should be justified accordingly (and the Introduction should be re-considered).
- It was concluded that the semi-professional football players are not heavily dependent on the aerobic energy system and this causes early decrease in performance and fatigue. There are a few issues here if the conclusion was based on the VO2max and the correlation coefficient to other measured variables.
First, the use of “VO2max” is not appropriate. A portable system was used to determine oxygen consumption during the YoYo test. However, there was no information given about whether/how maximum O2 consumption was achieved (e.g. criteria used). Normally, the test result can be regarded as VO2peak in a particular study, that may not represent the “true” maximum.
Secondly, there was no evidence in this study that supports the statement that these players are not heavily dependent on the aerobic system, and that this causes early fatigue. For a 90 min football match, surely it depends on the aerobic system for recovery during the low intensity intervals after sprints. For the two tests examined (~90 s and under 50 s respectively), aerobic system may not be the major energy source during the test but would be during the recovery intervals. The authors should clearly differentiate between these conditions in discussion on the contributions/importance of the aerobic system to football play (or to the test performance).
Thirdly, if the oxygen consumption and respiratory variables were not measured during the AST and CST tests, it is possibly not appropriate to correlate the VE and RER data, etc., from the YoYo test with the performance of the AST and CST, as they are test specific.
- Some specific comments
L161-167, have the authors considered corrections to the t test results for multiple comparisons?
L113, where is #3 in the diagram?
L142, something missing after the brackets (grammar).
L287, as mentioned above, this (conditions corresponding to those of a football game) should be better justified.
L318-322, as mentioned above, these statements need to be carefully justified.
L325-326, is this possible in a game?
Author Response
Comments and Suggestions for Authors
Comment 1
The work presented in this manuscript examined the performance of semi-professional football players in AST and CST. The authors concluded that the AST was more dependent on the anaerobic glycolytic system than CST, more performance decrement and fatigue were found in AST that CST, and the players were not heavily dependent on aerobic system that caused early decrease in performance and fatigue.
Although to examine the physical fitness profile of football players and develop optimal testing and training methods are important, and this study may have a merit if that was case, the manuscript as presented does not effectively achieve this goal and some of the conclusions do not appear to be supported by the data collected in this study. I would provide the following comments and critiques for authors’ consideration, aiming to improve the quality of the presentation.
Response 1
We thank the reviewer for the time and effort in reviewing this manuscript and appreciate the constructive feedback. Below we have answered the comments and suggestions from the reviewer. We have marked changes to the text in green.
Comment 2
The rationale and aim of the study must be better justified.
As the authors stated in Lines 70-72, it is very difficult to mimic the movement of football players (not to mention their specific positions) during a football match play by laboratory or field tests. Therefore, it would be better to provide more detailed background information, in physiological terms, about the tests to be examined. E.g., what are previous findings in respect the general sprint and low intensity movement durations, and evidence of fatigue or rate of fatigue, during a football match, and whether/how the tests used in this study could simulate the football profile (or not). The authors appeared to trying to explain the physiological/metabolic characteristics of football players in the Introduction, but failed to effectively link these to the tests to be used. If the aim was to assess fatigue in these tests, it should be justified accordingly (and the Introduction should be re-considered).
Response 2
Playing football, regardless of the sports level, is based on moving with high intensity, which is separated by intervals of low and medium intensity activities (Burgess, Naughton and Norton, 2006; Haugen, Tonnessen, Hisdal & Seiler, 2013; Rampini, Coutts, Castagna, Sassi and Impellizzeri, 2007). Accelerations in a match lasting from 2-4 s (Barnes et al., 2014) occur many times in succession, often not separated by a clear break (Bloomfield, Polman and O'Donoghue, 2007; Rampini et al., 2007, Brughelli, Cronin, Levin and Chaouachi, 2008). The importance of intense efforts is given by the fact that they are the ones that determine the result in a match (Faude, Koch, & Meyer, 2012; Mohr, Krustrup, & Bangsbo, 2003). In football, equally important, and often more important, is the ability to accelerate non-linearly (Cardoso de Araújo, Baumgart, Freiwald & Hoppe, 2017). About 85% of maximum speed maneuvers consist of curvilinear sprints of varying curvature (Bloomfield et al., 2007, Caldbeck, 2019 Smith, Dyson, Hale and Janaway, 2006; Smith, Dyson and Hate, 1997). The curves of the run usually have a radius of 3.5-11 m and are performed by players in all positions, except for goalkeepers (Caldbeck, 2019). The assessment of this area of the player's preparation, with particular emphasis on working in conditions of increasing fatigue, is important in planning the training process. AST and SCT tests require multiple direction changes simulating the movement of a soccer player during the game. On the other hand, multiple use makes it possible to bring the subject's assessment closer to the conditions of the game, during which this effort is performed repeatedly.
Comment 3
It was concluded that the semi-professional football players are not heavily dependent on the aerobic energy system and this causes early decrease in performance and fatigue. There are a few issues here if the conclusion was based on the VO2max and the correlation coefficient to other measured variables.
Response 3
This conclusion was not based on VO2max values. It results from the size of the LA gain after both specific Ajax and Curved tests. In the second repetition (series), the time of performing the effort with maximum intensity does not decrease significantly. At the same time, the increase in the anaerobic glycolysis metabolite - lactate is minimal. Thus, the energy needed to perform the work in the second series comes from aerobic metabolism, which results from the low glycolytic capacity of the subjects. This is typical of players who do not train using interval training and are not adapted to variable and high-intensity work.
Comment 4
First, the use of “VO2max” is not appropriate. A portable system was used to determine oxygen consumption during the YoYo test. However, there was no information given about whether/how maximum O2 consumption was achieved (e.g. criteria used). Normally, the test result can be regarded as VO2peak in a particular study, that may not represent the “true” maximum.
Response 4:
Getting the "real" maximum in a field study is always uncertain. The suitability of the values determined in laboratory tests for training practice is also debatable. Studies by CMWELLS, AM EDWARDS, EM WINTER, ML FYSH, B. DRUST Sport-specific fitness testing differentiates professional from amateur soccer players where VO2max and VO2 kinetics do no J SPORTS MED PHYS FITNESS 2012; 52: 245-54 show that in In the Yo-Yo test, the VO2max values are more realistic with respect to laboratory tests in semi-professionals compared to professionals. The anaerobic energy component contributes significantly to the result in the Yo-Yo test. If the competitor's disposition in this case is high, the result expressed as distance may be unreliable in relation to the VO2max value. However, if we conduct a direct measurement and the last stages would not lead to an increase in VO2, it is justified to consider the obtained value as the maximum value during running effort carried out in a test of gradually increasing intensity. This is because in each of the analyzed cases, the increase in work intensity during the last or penultimate exercise stage did not cause a further increase in VO2. If we assume that the plato for VO2 must persist for a longer time, then due to the time of the exercise stage in the Yo-Yo test, this condition may not be met. Some of the steps have the same intensity, which brings the test closer to tests with extended working time at the same intensity. VO2peak is the maximum (instantaneous) value recorded once. In our opinion, the values provided from direct registration corresponded to the VO2max criteria to a greater extent than the VO2peak criteria. Of course, the attention is very important and applies to all field studies where direct measurement of physiological parameters is used.
Comment 5
Secondly, there was no evidence in this study that supports the statement that these players are not heavily dependent on the aerobic system, and that this causes early fatigue. For a 90 min football match, surely it depends on the aerobic system for recovery during the low intensity intervals after sprints. For the two tests examined (~90 s and under 50 s respectively), aerobic system may not be the major energy source during the test but would be during the recovery intervals. The authors should clearly differentiate between these conditions in discussion on the contributions/importance of the aerobic system to football play (or to the test performance).
Response 5
It is absolutely impossible to say that the aerobic system is not important in securing the energetic effort. The study shows that at the semi-professional level, similar work intensity (in both 90 s and 50 s efforts) can be easily secured by the anaerobic and aerobic system as evidenced by a high increase in lactate after 1 series and a slight increase in blood lactate concentration in the 2 series despite the difference in work intensity.
Comment 6
Thirdly, if the oxygen consumption and respiratory variables were not measured during the AST and CST tests, it is possibly not appropriate to correlate the VE and RER data, etc., from the YoYo test with the performance of the AST and CST, as they are test specific.
Response 6
The correlation of the physiological parameters of the YoYo test with the parameters of the AST and CST tests was aimed at determining the relationship between the indicators of aerobic metabolism and work efficiency, assuming an anaerobic nature. The reason for adopting such a solution was the lack of reduction in work intensity during 2 series of both efforts. Despite the lack of activation of anaerobic metabolism, expressed by the lack of lactate gain after the second series, the time values were similar.
Some specific comments
Comment 7
L161-167, have the authors considered corrections to the t test results for multiple comparisons?
Response 7
We applied the Bonferroni correction for multiple comparisons
Comment 8
L113, where is #3 in the diagram?
Response 8
We would like to thank the reviewer for paying attention - we have made an appropriate correction
Comment 9
L142, something missing after the brackets (grammar).
Response 9
We thank the reviewer for bringing this to our attention. We have carefully proof-read the revised manuscript to eliminate grammatical errors.
Comment 10
L287, as mentioned above, this (conditions corresponding to those of a football game) should be better justified.
Response 10
Reduction of running efficiency during a match is a phenomenon that very often affects the sports result, and is the result of fatigue [Sarmento et al. 2014, Saunders et al. 2004,,. Burgess et al. 2010,]. The economy of movement is a determinant of endurance, which is a complex resultant of many functions, including metabolic and cardiopulmonary [Barnes et al. 2015, Saunders et al. 2004]. Metabolic efficiency relates to the use of available energy, while cardiorespiratory efficiency relates to the supply and use of oxygen in the working muscles. It is therefore related to heart rate, minute ventilation and VO2max. In football, which is a sport with a long duration and variable intensity, the movement economy is a relevant parameter of aerobic fitness. (Dolci et al. 2020). The credibility of the assessment of running economics requires the use of tests based on sport-specific movement patterns specific to football (Buchheit at al. 2011). The classic approach based on running sections in a straight line is not informative for football (Ziogas et al. 2011, Helgerud et al. 2002, Impellizzeri et al. 2006).
It is known that changes in direction are significantly more stressful than linear runs, causing higher heart rate (HR), blood lactate ([La]. (Buchheit et al. 2010, Dellal et al. 2010, Osgnach at al. 2010, diPrampero et al. . 2005).
Comment 11
L318-322, as mentioned above, these statements need to be carefully justified.
Response 11
In the research of Bradley et. al. (2009) in the Premier League showed that the recovery time between repetitions of high-intensity work increases. Between the 1st and the 6th of the 15-minute episode of the game, the cooldown time was 28% longer regardless of the position. Not only was a reduction in the number of accelerations observed, but also the frequency of attacks. Thus, the effectiveness of the game largely depends on the speed of regeneration during the match and the ability to work at maximum intensity throughout the match. This may support training to increase aerobic capacity, as Buchheit (2009) points out in his research on handball players. Training at the VO2max level can be effective in this respect by using intermittent efforts (Billat 2001, Midgley, Mc Naughton 2006, Midgley et al. 2006-7). The results of the research (Helgerud J et al. 2007) showed that the use of high intensity intermittent exercise in training may result in a greater improvement in maximum aerobic capacity and endurance than continuous exercise at an intensity of 85% of the maximum heart rate.
Comment 12
L325-326, is this possible in a game?
Response 12
Based on T. Reilly's scientific report, Energetics of high-intensity exercise (soccer) with particular reference to fatigue. Journal of Sports Sciences Volume 15, 1997 - Issue 3 pages 257-263 https://doi.org/10.1080/026404197367263 „Soccer entails intermittent exercise with bouts of short, intense activity punctuating longer periods of low-level, moderate-intensity exercise. High levels of blood lactate may sometimes be observed during a match but the active recovery periods at submaximal exercise levels allow for its removal on a continual basis. While anaerobic efforts are evident in activity with the ball and shadowing fast-moving opponents, the largest strain is placed on aerobic metabolism.”
Round 2
Reviewer 4 Report
The revised version of the manuscript has shown some improvements, and I appreciate authors’ efforts. However, I don’t think that the authors have adequately addressed all the issues raised. There are still several areas in the manuscript that require clarification and revision, as listed below.
- In respect of the Introduction, the revised aim of the study is clearer and better justifiable according to what was conducted in this study. In relation to the deficiency in the background information, the authors cited a number of references in their Response 2 (to the reviewer) to support the rationale of the study, however, none of these references is cited in the manuscript (except Mohr et al. 2003 that is cited in a later part of the article). The readers will not be able to understand these arguments if they are not presented in the article.
- In respect of the third conclusion (as in the Abstract) “(3) semi-professional football players are not heavily dependent on the aerobic energy system and it causes early decrease in performance and fatigue”, it is more of a speculation rather than a conclusion. The authors should be more cautious in making such a statement, as the results from this study are specific to the two tests examined, and there was no data collected from athletes at other levels, or from those of the same level in other studies, for comparison. The authors argued in their Response 3 that “……This is typical of players who do not train using interval training……”, but they have not collected evidence from this study (or cited references) to support a generalization of the conclusion 3. I would suggest authors add “in the AST and CST tests” to be more specific.
There is also a grammar issue, as it is not clear what the “it” refers to, e.g. the aerobic energy system, the tests examined, or football match?
- In respect of VO2max vs VO2peak, there are some more recent debates on this issue (e.g. Poole and Jones 2017, Green and Askew 2018, both in Journal of Applied Physiology). Obviously this study is not to address this issue. It is good to see that the authors have revised the statement on Line 114 to “to predict VO2max”, as the “true” VO2max could not be measured or justified in the YoYo test. However, the second half of the statement (Line 115-116) needs some further clarification. Were the VE, HR, RF, RER and O2 pulse values obtained (as measured) at the end of the test, or a prediction method was used to predict these variables at the “level of the predicted VO2max”? If so, how? The bottom line is that, any reader can follow your description to repeat your experiment.
- In authors’ Responses 10, 11, 12, again, most of the references cited in the arguments are not cited in the manuscript, if the authors would use these references to support their discussion.
- The conclusion statements at the end of the manuscript should be presented in a similar way as they are in the Abstract (e.g. 1,2,3). And again, in Lines 369-370, the statement about the football game is not supported by the data collected from this study, although it should be specific to the AST and CST.
- Line 188, 191: If the Bonferroni correction was applied for multiple comparisons, why P=0.05 is still used as the threshold of significance?
Some minor issues
Line 36: consider “…are more obvious in…” if it is not statistically significant.
Lines 44 and 47: there is no need to use hyphen “-“
Line 108: should it be “This test is a widely used…”?
Line 121-128: The description on the two sets of tests should be presented before Line 121, so the reader can understand what “both” refers to.
The “VO2max” used in the table titles as well as in Lines 195, 279 should be “predicted VO2max”.
Author Response
Dear Madam or Sir,
We thank the reviewer for the time and effort in reviewing this manuscript and appreciate the constructive feedback. Below we have answered the comments and suggestions from the reviewer.
Comments 1
In respect of the Introduction, the revised aim of the study is clearer and better justifiable according to what was conducted in this study. In relation to the deficiency in the background information, the authors cited a number of references in their Response 2 (to the reviewer) to support the rationale of the study, however, none of these references is cited in the manuscript (except Mohr et al. 2003 that is cited in a later part of the article). The readers will not be able to understand these arguments if they are not presented in the article.
Response 1
We absolutely agree with the reviewer. We have supplemented this section and included the relevant references.
Comments 2
In respect of the third conclusion (as in the Abstract) “(3) semi-professional football players are not heavily dependent on the aerobic energy system and it causes early decrease in performance and fatigue”, it is more of a speculation rather than a conclusion. The authors should be more cautious in making such a statement, as the results from this study are specific to the two tests examined, and there was no data collected from athletes at other levels, or from those of the same level in other studies, for comparison. The authors argued in their Response 3 that “……This is typical of players who do not train using interval training……”, but they have not collected evidence from this study (or cited references) to support a generalization of the conclusion 3. I would suggest authors add “in the AST and CST tests” to be more specific.
Comments 3
There is also a grammar issue, as it is not clear what the “it” refers to, e.g. the aerobic energy system, the tests examined, or football match?
Response 2 - 3
A very correct point, we fully agree with the reviewer. As suggested, we deleted sentence: “semi-professional football players are not heavily dependent on the aerobic energy system and it causes early decrease in performance and fatigue.” and add this: “Early decrease in performance and fatigue in the semi-professional football players in AST and CST may be due to the insufficiency of their aerobic energy system.”
Comments 4
In respect of VO2max vs VO2peak, there are some more recent debates on this issue (e.g. Poole and Jones 2017, Green and Askew 2018, both in Journal of Applied Physiology). Obviously this study is not to address this issue. It is good to see that the authors have revised the statement on Line 114 to “to predict VO2max”, as the “true” VO2max could not be measured or justified in the YoYo test. However, the second half of the statement (Line 115-116) needs some further clarification. Were the VE, HR, RF, RER and O2 pulse values obtained (as measured) at the end of the test, or a prediction method was used to predict these variables at the “level of the predicted VO2max”? If so, how? The bottom line is that, any reader can follow your description to repeat your experiment.
Response 4
Thank you for paying attention. We revised sentence from lines 114-116. We made a correction to the entire sentence. It now reads as follows:
“The purpose of this test was to predict VO2max and measure variables like pulmonary ventilation (VE), heart rate (HR), respiratory frequency (RF), respiratory exchange ratio (RER) and oxygen pulse (VO2/beat), at the most exhaustive stage of the test (that likely indicates the attainment of the VO2max).”
Comments 5
In authors’ Responses 10, 11, 12, again, most of the references cited in the arguments are not cited in the manuscript, if the authors would use these references to support their discussion.
Response 5
We absolutely agree with the reviewer. We have supplemented this section and included the relevant references.
Comments 6
L318-322, as mentioned above, these statements need to be carefully justified.
Response 6
Again, we absolutely agree with the reviewer. We have supplemented this section and included the relevant references.
Comments 7
L325-326, is this possible in a game?
Response 7
We based this statement on an article by Thomas Reilly.
- Reilly, Energetics of high-intensity exercise (soccer) with particular reference to fatigue. Journal of Sports Sciences Volume 15, 1997 - Issue 3 pages 257-263 https://doi.org/10.1080/026404197367263
„Soccer entails intermittent exercise with bouts of short, intense activity punctuating longer periods of low-level, moderate-intensity exercise. High levels of blood lactate may sometimes be observed during a match but the active recovery periods at submaximal exercise levels allow for its removal on a continual basis. While anaerobic efforts are evident in activity with the ball and shadowing fast-moving opponents, the largest strain is placed on aerobic metabolism.”
Comments 8
The conclusion statements at the end of the manuscript should be presented in a similar way as they are in the Abstract (e.g. 1,2,3). And again, in Lines 369-370, the statement about the football game is not supported by the data collected from this study, although it should be specific to the AST and CST.
Response 8
Thank you for this suggestion, we have simplified this part of the manuscript as much as possible, keeping the content consistent with the abstract.
Comments 9
Line 188, 191: If the Bonferroni correction was applied for multiple comparisons, why P=0.05 is still used as the threshold of significance?
Response 9
Thank you for paying attention. We have introduced a corresponding explanation of the Bonferroni correction to the statistical analysis part. Introducing the Bonferroni correction, the significance level for multiple comparisons is p = 0.008
Comments 10
Line 36: consider “…are more obvious in…” if it is not statistically significant.
Response 10
Thank you for paying attention. We have made modifications to the text.
Comments 11
Lines 44 and 47: there is no need to use hyphen “-“
Response 11
Thank you for paying attention. We have made modifications to the text.
Comments 12
Line 108: should it be “This test is a widely used…”?
Response 12
Thank you for paying attention. We made an appropriate correction.
Comments 13
Line 121-128: The description on the two sets of tests should be presented before Line 121, so the reader can understand what “both” refers to.
Response 13
Thank you for a very good suggestion. We made an appropriate correction.
Comments 14
The “VO2max” used in the table titles as well as in Lines 195, 279 should be “predicted VO2max”.
Response 14
Thank you for paying attention. We made an appropriate correction.
